

# Southwestward propagating quasi-biweekly oscillations over the South-West Indian Ocean during boreal winter: Characteristics and propagation mechanism

Sambrita Ghatak[1,2,*] and Jai Sukhatme[1,2,*]

[1]Centre for Atmospheric and Oceanic Sciences, Indian Institute of Science, Bangalore, India.
[2]Divecha Centre for Climate Change, Indian Institute of Science, Bangalore, India.
[*]These authors contributed equally to this work.

**Correspondence:** Sambrita Ghatak (pulu.dec@gmail.com)

**Abstract.** An analysis of outgoing longwave radiation (OLR) and winds over the South-West Indian Ocean (SWIO) yields regular, poleward propagating, large-scale, convectively coupled systems of alternating cyclonic and anticyclonic circulation with a quasi-biweekly period during boreal winter. Composites from 30 years (1980/81 - 2009/10) of OLR and reanalysis data show well-formed rotational gyres in the lower troposphere (700 hPa) that can be tracked from near the equator to almost 30°S appearing west of Sumatra and propagating towards Madagascar, i.e., with mean southwest propagation. The gyres show a marked northwest-southeast tilt, giving rise to a northeast-southwest oriented wavetrain. The scale of the gyres is about 30°-35°, their period is 14-18 days and they have a westward phase speed of approximately 3.5 ms$^{-1}$. The group velocity of these wave packets is near-zero. In early stages, the gyres are associated with weak convection, but when they move poleward and cross 10°S, convective coupling is enhanced. Velocity fields and OLR indicate that the maxima of moist convective activity lies in the eastern sector of the gyres and a comparison between column-integrated moisture and OLR anomalies shows they are highly collocated, indicating the applicability of the moisture mode framework. A moisture budget reveals that once the gyres reach 10°-20°S, moistening is mainly due to northerlies in the eastern sector of the cyclonic gyre acting on the meridional gradient of background moisture, which eventually gives rise to anomalous convection in this region. This moistening process continues up to 30°S while the gyres traverse southwestward. Subsequently, background easterlies advect anomalous moisture and along with moistening via so-called 'column-processes', convection is observed to extend inside the gyre from the eastern side. A vorticity budget reveals that the $\beta$ effect plays a leading role in the southwestward propagation, horizontal advection assists the westward movement of vorticity anomalies due to prevailing easterlies and moist coupling (via stretching) is important in reducing the speed of propagation of this mode. In fact, the relatively slow southwestward movement of the system is because moist coupling reduces the effect of $\beta$ and horizontal advection terms. Moreover, as convection primarily takes place on the eastern side of the vortex, and somewhat inside the vortex too, it also follows a southwestward path along with the QBWO vortices.



## 1 Introduction

The quasi-biweekly oscillation (QBWO), with a timescale between that of synoptic disturbances and the Madden-Julian Oscillation (MJO; Madden and Julian, 1971, 1972, 30-60 days), is an important component of tropical intraseasonal variability
(Kikuchi and Wang, 2009). Early reports of the QBWO were in the context of the Indian summer monsoon, with fluctuations of a 15-20 day period noted in lower tropospheric meridional wind anomalies (Keshavamurty, 1972). From then on, features, mechanisms, and influence of the QBWO has been studied over many basins worldwide, particularly over the Indian Monsoon region (IMR), the western North Pacific (WNP), the South China Sea (SCS), and Indo-China (Krishnamurti and Bhalme, 1976; Krishnamurti and Ardanuy, 1980; Chen and Chen, 1993; Fukutomi and Yasunari, 1999; Chan et al., 2002; Chatterjee and
Goswami, 2004; Mao and Chan, 2005; Kikuchi and Wang, 2009; Chen and Sui, 2010; Jia and Yang, 2013; Wang and Chen, 2017). In their review paper, Kikuchi and Wang (2009) have documented westward propagating QBWO activity of tropical origin over several geographic locations around the tropics.

From a theoretical perspective, QBWOs of tropical origin, particularly those which propagate westward have been interpreted as gravest meridional mode equatorial Rossby (ER) waves, modified by mean flow and convective coupling (Chatterjee and
Goswami, 2004; Kikuchi and Wang, 2009; Chen and Sui, 2010). Thus, here we use the term convectively coupled ER (CCER) waves and QBWO interchangeably. Though the structural features of QBWO/CCER waves have some similarities with the ER waves from dry shallow water theory (Matsuno, 1966), there are some very important differences. Particularly, its phase speed is quite slow when compared to the dry ER waves (Wheeler et al., 2000; Wheeler and Kiladis, 1999; Chatterjee and Goswami, 2004; Kiladis et al., 2009; Chen and Sui, 2010), and the position of strongest convection is also not consistent with dry ER
wave solutions (Chatterjee and Goswami, 2004; Chen, 2022; Wang et al., 2017). Moreover, in some ocean basins, unlike the theoretical dry waves, these modes also show prominent poleward propagation (Molinari et al., 2007; Wang and Chen, 2017; Chen and Sui, 2010). Traditionally, the slower phase speed is understood as an effect of latent heat release by the associated convection, which reduces the equivalent depth (Kiladis et al., 2009). There are few variations in this school of thought, for example, via an idealized model study, the influence of boundary layer dynamics was considered, and it was suggested that
unstable ER waves, mainly driven by convective feedback via frictional boundary level convergence and modulated by the mean flow account for the observed structure of the QBWO (Chatterjee and Goswami, 2004). Fundamentally, as this mode is convectively coupled, its interaction with moisture is of particular interest. Though traditional views of convectively coupled ER waves accounted for the impact of latent heat released by precipitation (Chatterjee and Goswami, 2004; Kiladis et al., 2009), they didn't consider prognostic moisture, thus the possible importance of moisture variations in the dynamics of this
system was ignored. Even though these traditional efforts has some success in producing solutions that propagate westward with a phase speed close to the observed QBWO, there are many inconsistencies with respect to observations (Mayta et al., 2022; Chen, 2022).

With the advent of moisture mode perspective (Sobel et al., 2001; Sobel and Maloney, 2013; Sukhatme, 2014; Adames and Kim, 2016; Kim et al., 2014; Kiranmayi and Maloney, 2011), these traditional views are seen with growing skepticism. In the





moisture mode view, anomalous convection is primarily dictated by the column-moisture anomaly. In other words, anomalous convection follows the evolving moisture anomaly, thus prognostic moisture is essential to capture these modes in simplified models. Till date, this perspective has found use in the context of the 30-60 day period MJO/BSISO, and now it is widely accepted that interactive moisture forms an essential piece in the dynamics of these modes (Benedict and Randall, 2007; Maloney, 2009; Kiranmayi and Maloney, 2011; Adames and Kim, 2016) - in fact, aquaplanet model experiments show that MJO-like modes disappear when moisture becomes dynamically passive (Suhas et al., 2022). Specifically, the moisture mode paradigm has shown significant promise in uncovering the fundamental dynamics, structure and the mechanism of eastward propagation of the MJO (Kiranmayi and Maloney, 2011; Kim et al., 2014; Adames and Wallace, 2015; Adames and Kim, 2016; Zhang et al., 2020; Gonzalez and Jiang, 2019) and northward propagation of the BSISO (Jiang et al., 2018; Ghatak and Sukhatme, 2024; Chen and Wang, 2021).

Recent observational studies have found that the QBWO also exhibits a strong coherence between column-moisture and precipitation, suggesting that it too can be viewed through the lens of a moisture mode (Gonzalez and Jiang, 2019; Mayta et al., 2022). Following this approach, moisture/moist static energy budget analyses have been performed to understand the evolution of moisture associated with this mode over different ocean basins (Li et al., 2020; Mayta et al., 2022; Mayta and Adames Corraliza, 2024; Dong et al., 2024). Indeed, simplified models with prognostic moisture also suggest that westward propagating modes with some resemblance with the observed QBWOs can emerge as moisture mode solutions (Sukhatme, 2014; Fuchs-Stone et al., 2019; Mayta et al., 2022). Here, Fuchs-Stone et al. (2019) obtained westward propagating moisture modes which critically depend on wind-induced surface heat exchange (WISHE), while Sukhatme (2014) and Mayta et al. (2022) obtained westward propagating moisture modes with imposed horizontal moisture gradients, though Mayta et al. (2022) also considered critical importance of background easterlies in their model. Though the application of moisture mode perspective is growing in QBWO studies, our understanding of the role of moisture variation on this mode is far from conclusive.

Another aspect of the QBWO which is shrouded in mystery is its pronounced poleward propagation (along with westward propagation) in some basins. Most theoretical treatments discussed above tried to capture the slow westward propagation of the QBWO, as they mainly focused on the tropical Pacific or the Indian Monsoon Region QBWO/moist ER waves, which primarily move westward. In literature, QBWOs with poleward propagating components are mostly discussed in the Western North Pacific or South China Sea basin (Chen and Sui, 2010; Wang et al., 2017; Li et al., 2020), though there is no consensus regarding the mechanism that drives their poleward movement. Early studies suggested that this propagation might be due to interaction between ER waves and the prevailing mean flow (Kikuchi and Wang, 2009; Chen and Sui, 2010), while recent work has suggested the possible role of moisture convergence, horizontal moisture advection and air-sea interaction (Li et al., 2020; Dong et al., 2024).

In this study, we will focus on South-West Indian Ocean (SWIO) basin, where the existence of a poleward propagating QBWO has been noted, but has not been examined in detail. In the Indian Ocean region, most of the QBWO related work has focused on the boreal summer in the context of Indian Summer Monsoon, though, a few early studies also noted wind fluctuations on the quasi-biweekly time scale and their influence on the central and SWIO during boreal winter (Sengupta et al., 2004;



Han et al., 2007). Indeed, the existence of convective disturbances with a similar timescale in this basin has been noted in
some form or the other (Jury and Pathack, 1991; Jury et al., 1991; Bessafi and Wheeler, 2006; Kikuchi and Wang, 2009), but
detailed analyses of the QBWO in this basin are very limited, and its movement, structure, characteristics and propagation
mechanism remained mostly unexplored. Until recently, the work of Fukutomi and Yasunari (2013, 2014) comprised the most
detailed examination in this basin regarding sub-monthly convectively coupled disturbances. These studies showed regular
southwestward propagating wavetrains, but their choice of a broad filter (6-30 day) might have mixed various disturbances
with different temporal and spatial scales and characteristics, such as ER and mixed Rossby-Gravity waves as well as synoptic
disturbances. Very recently, a couple of studies focused particularly on the QBWO timescales (14-30 days and 10-20 days
respectively), and both of them identified a south-westward moving QBWO mode with a period around 18 days in this basin
during boreal winter (Ghatak and Sukhatme, 2021; Yang et al., 2023). While Ghatak and Sukhatme (2021) studied its vorticity
dynamics, Yang et al. (2023) focused on its moisture dynamics, but still the holistic mechanism of the propagation of this
convectively coupled system is missing. Particularly, how the convection couples with the circulation is not very clear. In this
paper, we focus on these aspects, as these might help us to unravel the moist dynamics of QBWO, and thus the mechanism of
slow westward and poleward propagation in particular.

A more practical significance of this mode in the SWIO is related to its possible influence on tropical cyclones (TCs) during
the boreal winter. In fact, the influence of QBWO on tropical cyclogenesis is well documented in other ocean basins (Ling
et al., 2016; Zhao et al., 2016). In the SWIO, on average, the number of TCs formed each year in this basin ranges from 9-10 to
12-13 depending on the period considered (Mavume et al., 2009; Muthige et al., 2018; Leroux et al., 2018). A majority of these
storms develop during the boreal winter and affect Madagascar, other SWIO islands, and sometimes south-east African nations,
causing widespread devastation (Vitart et al., 2003; Reason and Keibel, 2004; Fitchett and Grab, 2014; Leroux et al., 2018;
Bousquet et al., 2021). Currently, the track and strength of TCs in SWIO are difficult to predict accurately more than a few days
ahead (Kolstad, 2021), making it hard to provide the time required for evacuation and other preparedness measures (Webster,
2013; White et al., 2017; Mavhura, 2020). Bessafi and Wheeler (2006) also found a statistically significant relationship between
convectively coupled equatorial Rossby (CCER) waves and cyclogenesis in this basin, and this CCER wave might well be
connected with the QBWO. Given the 8-12 day predictability horizon for these modes (Li and Stechmann, 2020), this signal
can be exploited for advance prediction of TCs. Thus, a better understanding of the QBWO in this region is relevant for
improved forecasts of cyclones in the SWIO basin.

Apart from influencing extreme events such as TCs, the QBWO also has a bearing on regional rainfall patterns. Specifically, a
10-20 day cycle in boreal winter rainfall is documented in Madagascar (Nassor and Jury, 1998). Westward propagating transient
waves of similar timescale have been observed to contribute to fluctuations in rainfall in the south-east African coastal region,
and to cause heavy rainfall episodes even in the interior of the continent, such as the Southern African Plateau (Jury and
Pathack, 1991; Jury et al., 1991). It is quite possible that QBWO has a role to play in these observed rainfall events. In turn, this
provides a valuable framework for active and break cycle prediction in Madagascar and south-east Africa that predominantly





depends on rain-fed agriculture (Malherbe et al., 2012; Silva and Matyas, 2014; Macron et al., 2016; Pohl et al., 2017; Rapolaki et al., 2019).

In this paper; after a description of the data and methods used in the study (Section 2), we proceed with a detailed documentation

of the characteristic features of the QBWO in the SWIO (Section 3), including its spatial and temporal scales, propagation, horizontal structure and evolution. Our analysis is based on composites generated from thirty years of daily satellite and reanalysis data. With the characterisation in hand, we then proceed to a detailed moisture budget (Section 4) to understand the propagation of convection associated with this mode. Then we proceed to a vorticity budget in section 5 to understand propagation mechanism of the QBWO vortex, and in section 6, we present discussion ans conclusion, where we discuss the

mechanism how moisture and circulation combines to give rise to the propagation characteristics, and its possible implications.

## 2  Data and Methodology

Four-times daily products from ERA5 reanalysis project serve as the main data set for this study. We have calculated daily mean values of meteorological variables from this data. Specifically, we have used 30 years of horizontal winds, specific humidity, and vertical velocity data at 17 pressure levels (1000 to 200 hPa with an interval of 50 hPa) and evaporation and precipitation

rate at single level that spans December to March (DJFM) from 1980/1981-2009/2010 and has a horizontal resolution of 2.5°. This data is used to calculate the derived fields presented in this paper. Some fields, such as relative vorticity and various terms of the vorticity and moisture budgets are computed using Windspharm package (Dawson, 2016). Daily outgoing longwave radiation (OLR) data of a spatial resolution of 2.5° from the National Oceanic and Atmospheric Administration (NOAA) satellites serves as a proxy for moist convection (Liebmann and Smith, 1996).

To isolate the QBWO signal, we used a filter with a 10-25 day band. Most studies regarding QBWO use a 10-20 day filter (Chatterjee and Goswami, 2004; Chen and Sui, 2010; Wang et al., 2017), but there are variations, such as 12-20 days (Kikuchi and Wang, 2009; Jia and Yang, 2013), 10-25 days (Fukutomi and Yasunari, 1999), and 12-24 days (Chen and Chen, 1995). Here, we use a lower limit of 10 days to exclude synoptic scale signals, and the upper limit of 25 days is used instead of widely used 20 day as previous studies have shown that there can be systems of QBWO scale which have a slightly larger period than

20 days (Molinari et al., 2007). We note that significant power associated with MJO is well above 30 day, so we don't run into the risk of including MJO signals while using a 10-25 day band window.

To extract 10-25 day variability, a Lanczos band-pass filtering method is used (Duchon, 1979). Further, to highlight the QBWO, the annual cycle of individual years are removed by subtracting the mean and the first three Fourier harmonics before filtering is done. To check robustness, we varied the filtering band and it produces essentially identical results up to a 10-30 day range,

so our result is not sensitive to the width of the filter as long as it satisfactorily isolates QBWO signal from the variability of other tropical modes. To describe the structure and evolution of the QBWO, we have constructed composites of OLR and lower free-tropospheric (700 hPa) circulation from the filtered data. Members of the composites satisfy the following criteria: 1) In the chosen box (55°-62.5° E, 10°-17.5° S; Figure 1); a region of high 10-25 day variability (in terms of standard deviation) of



OLR, the negative anomaly of box-averaged OLR on Day 0 (which is the day when the box-averaged OLR is locally minimum) must be less than -1 standard deviation; 2) The identified minima should be outside the first and last 10 days of the season, so that there is room for a complete cycle of 20 days within the season (DJFM).

These criteria result in 109 members from 1980/81-2009/10 which are then used to make the composites. We also varied the base region to check whether the results depend strongly on the particular box chosen. Indeed, as long as the base box remains inside the region of high filtered OLR standard deviation, the composite structure is not affected. Further, the composite vorticity and moisture budgets of these systems are also analyzed to understand the evolution of the QBWO and its propagation. A detailed description of these analysis methods are given in the respective sections. For all the constructed composites, we have performed student's-t significance testing, OLR and winds are shown only when they are significantly different from zero at 95% confidence level, while for the budget related calculations, we have plotted the full signal, but we confirm that all the signals at almost all the pixels in the location of interest are statistically significant at 95% confidence level.

## 3 Boreal Winter Quasi-biweekly Activity in SWIO

The winter season (DJFM) 10-25 day variability of organized moist convection in terms of the standard deviation of filtered OLR is shown in Figure 1. The north-east to south-west (NE-SW) tilt of moist convective activity in the southern Indian Ocean suggests that this could be a pathway for the propagation of quasi-biweekly intraseasonal systems in this region. Particularly, we find a patch of heightened convective activity in the SWIO east of Madagascar, which suggests that it might be an epicentre of the QBWO in this region. Thus, we take our box to construct composites inside this area of largest quasi-biweekly convective activity (blue square in Figure 1). Interestingly, the band of heightened convective activity extends towards the west into the continent of Africa along 20°S. This suggests that QBWO activity over the SWIO might stretch into Africa, or this might be a region of independent quasi-biweekly activity. We will explore these possibilities in the following sections.

### 3.1 Horizontal Structure

Composites of the southwestward moving intraseasonal systems are presented in Figure 2 in terms of horizontal wind at lower-free troposphere (700 hPa) and OLR. On Day −8, northeast of Madagascar, we see a well-developed anticyclonic circulation coupled with positive OLR anomalies (red color in Figure 2) indicating reduced moist convection. Meanwhile, near the equator, we see an emerging low between 50°-80°E with weak negative OLR anomalies. By Day −5, the newly formed cyclonic vortex near the equator becomes more prominent and moved slightly westward, and associated OLR anomalies become little stronger. This newly formed vortex has a northeast-southwest (NE-SW) tilt. Simultaneously, near Madagascar, the anticyclonic circulation and associated positive OLR anomalies have moved to the southwest, expanded but become weaker and made some inroads into south-east Africa, especially parts of the coast and the Great Rift Valley. In effect, a clear NE-SW oriented wavetrain pattern has been established.



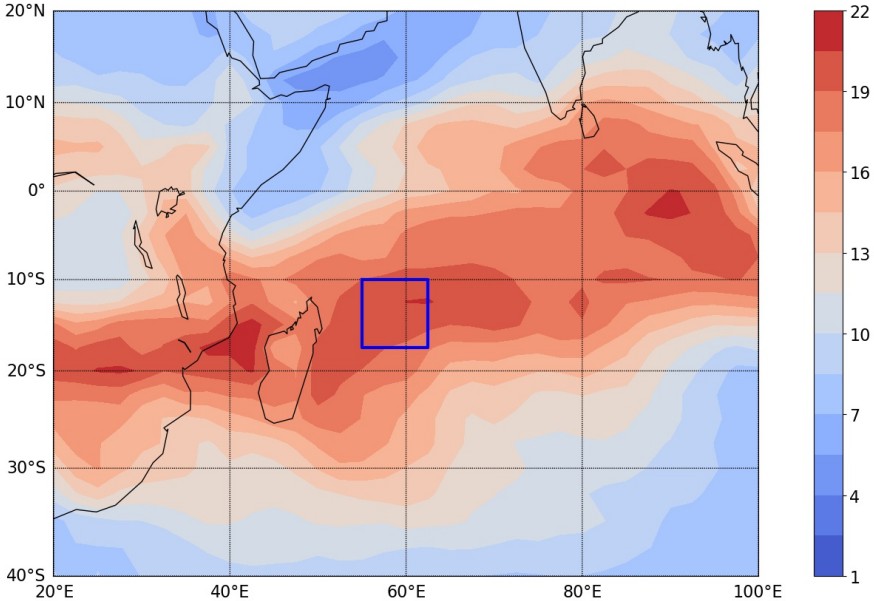

**Figure 1.** Geographical distribution of the standard deviation of 10-25 day filtered OLR for the boreal winter (DJFM) during the period 1980/1981-2009/10. The blue box is the region chosen for composite analysis. The unit for the standard deviation of OLR is W m$^{-2}$

Moving ahead, by next 3 days, from Day $-4$ to Day $-2$, the cyclonic anomaly moved southwestward, and a dramatic strength-

ening of convection can be seen inside the gyre, especially in the right (eastern) part of the gyre, in the vicinity of the northerly
flows. This southwest movement of the convectively coupled vortex continues and by Day $0$, the cyclonic gyre covers most of
Madagascar. The anomalous convection increases in strength and also moves southwest compared to Day $-2$, now it extends
into the gyre from the eastern side, though eastern part of the gyre is still the location of maximum convection. The tilt of the
gyre has reduced, and it is more zonally oriented at this stage. In succession, near the equator, we now see the hint of the birth of

the next high. In the next 3 days, the cyclonic gyre continue moving southwestward, and the associated convection extends into
south-east Africa on one hand, though with weak magnitude, and south of 20°S on the other, thus forms a bow-like structure.
Overall, the convection starts getting weaker. An area of suppressed convection with anticyclonic flow starts forming near the
equator to the north-east of the cyclonic vortex, thus again, a NE-SW oriented wavetrain is clearly visible. By inspection, we
can say, that the oscillation approximately completes a half cycle in 7-9 days, thus it has a time period around 14-18 days.

Interestingly, the QBWO in the SWIO basin doesn't extend deep into southern Africa, thus, the high OLR standard deviation
in the QBWO scale in that region (Figure 1) is probably due to some other independent disturbance with similar timescale,
possibly an eastward propagating mode as described by Yang et al. (2023).

Overall, the low level (850 hPa) circulation as well as convection anomalies exhibit southwestward propagation of the QBWO
over the SWIO. The vortex starts near the equator between roughly 50°-90°E with weak convective anomalies, gradually

moves southwestward and becomes convectively coupled, preferring the eastern side of the vortex, continues southwestward




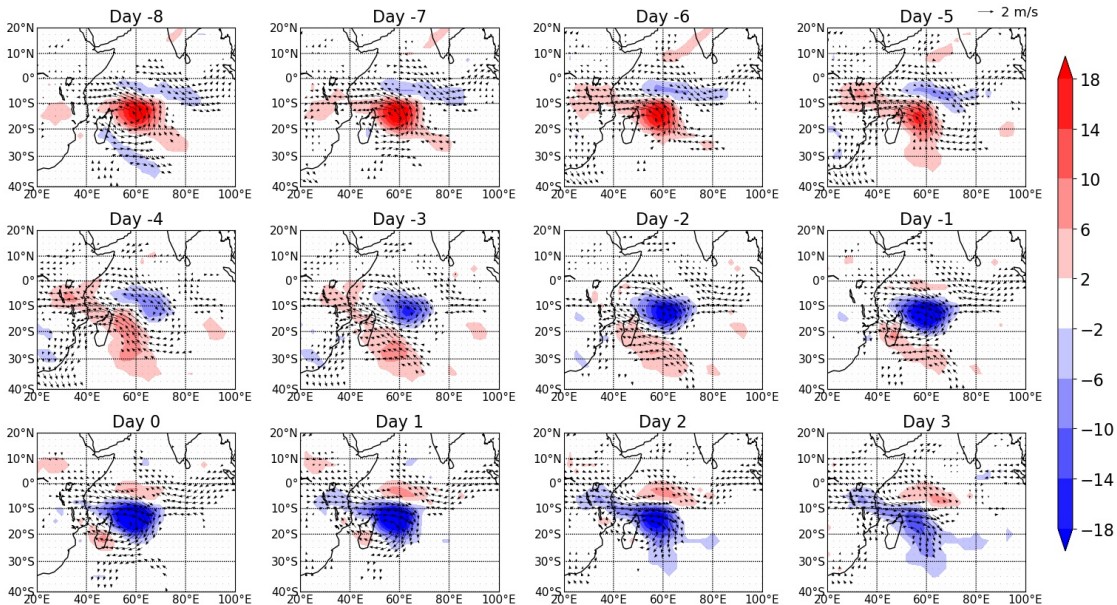

**Figure 2.** Composite of 10-25 day filtered OLR (W m$^{-2}$; shading) and 700 hPa wind anomalies (quivers) from Day $-8$ to Day 3. OLR and wind vectors are statistically significant at 95% confidence level.

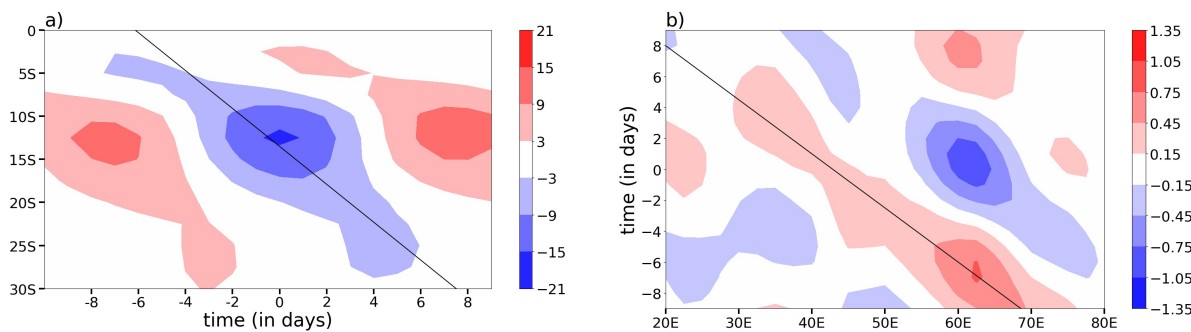

**Figure 3.** Latitude-time and longitude-time diagrams of 10-25 day filtered OLR (W m$^{-2}$) anomalies averaged over $40°$-$70°$E (Panel a) 10-25 day filtered meridional wind (m s$^{-1}$) anomalies averaged over $5°$-$25°$S (Panel b) from the composite in Figure 2. The black lines correspond to a southward movement of $2.2°$/day (Panel a) and $2.9°$/day (Panel b). Colors shown are statistically significant at 95% confidence level.

with the anomalous convection, and finally weakens after reaching Madagascar while the convection extends towards the south and the west. We note a clear northwest-southeast (NW-SE) tilt associated with its gyres, and alternating cyclonic and anticyclonic circulation patterns together give rise to a NE-SW oriented wavetrain. As the QBWO moves further away from the equator, the tilt of the gyre reduces. Finally, the oscillation dies down southeast of Madagascar, near $30°$S traversing a total meridional distance of about 20-25 degrees. Further, the gyre at a mature phase cover around 30-35 degrees of longitude while



the latitudinal extent is approximately 20 degrees. This suggests that the zonal wavelength of the system is roughly 6500-7500 km (or, wavenumber 5-6).

To estimate the period and speed of propagation of the QBWO, we now construct time-latitude and time-longitude Hovmöller diagrams. Here, we need to note that the vortex as well as the associated convection doesn't travel consistently towards one

direction, thus these are just rough estimates. Specifically, Figure 3a shows the time-latitude diagrams of filtered OLR and zonal wind anomalies averaged over 40°-70° E, depicting a southward phase speed of approximately 2.2°/day. Similarly, Figure 3b shows the longitude-time diagrams of filtered meridional wind anomalies averaged over slightly off the equator (5°-25° S) and these indicate a westward speed of about 2.9°/day or 3.5 m s$^{-1}$. The group velocity that can be estimated from 3a and 3b is quite close to zero. The wavelength that can be estimated here also matches with our estimation discussed above. The time

period estimated from both these figures is also consistent, i.e., approximately 14-18 days.

In all, our QBWO composite is somewhat similar to its boreal summer northern hemisphere counterpart documented in the SCS (Wang and Chen, 2017), where the QBWO is asymmetric about the equator, i.e., mostly confined to the summer hemisphere, and propagates northwestward. As mentioned, the QBWO is usually understood as a modified equatorial Rossby wave, and ER waves observed by Molinari et al. (2007) also have a similar hemispheric asymmetry and prominent poleward propagation.

Further, in the boreal winter, the area of very high specific humidity and its gradient is concentrated in the Southern Hemisphere in the Indian Ocean region, which points to the possibility of moist coupling and growth south of the equator. The wavelength (around 6500-7500 km), time-period (14-18 days) and westward phase speed (3.5 m/s) are also comparable to the range of estimates for CCER waves/QBWOs in various geographic locations (Numaguti, 1995; Wheeler and Kiladis, 1999; Bessafi and Wheeler, 2006; Wheeler et al., 2000; Janicot et al., 2010; Chatterjee and Goswami, 2004). This is also similar to the

QBWO documented by Ghatak and Sukhatme (2021) in SWIO, and has many similarities with the CCER wave documented by Bessafi and Wheeler (2006) in the same basin, though their waves have limited poleward propagation , probably due to their filtering process. Quite remarkably, the length and time scales, as well as phase speed observed here match with that of westward propagating equatorial modes in an idealized moist shallow water system (Sukhatme, 2014). The movement of wave packets (Figure 2) suggests a near-zero group velocity which also agrees with observations of CCER waves (Wheeler

and Kiladis, 1999; Wheeler et al., 2000; Molinari et al., 2007) and QBWOs (Chen and Sui, 2010; Wang and Chen, 2017). A distinguishing feature of the QBWO over SWIO is that the strongest convection - or, most negative OLR anomaly - is located towards the eastern side of the cyclonic vortex, aligning with the QBWO northerlies, though, in its mature phase, convection extends towards the centre of the vortex. We will find the cause of this behaviour in the moisture budget section.

Notably, these results are somewhat different than those presented by Fukutomi and Yasunari (2013) in their study of sub-

monthly intraseasonal activity in the SWIO. They found meridionally elongated cyclonic and anti-cyclonic vortices and documented continuous northeast-southwest propagation, whereas we observe an southwest movement of mostly zonally elongated gyres with some meridional tilt. Though the estimated phase speed in their study is close to ours, they noted an eastward group velocity of 8 m s$^{-1}$, which we clearly do not observe in our composite. These differences may be artefacts of using different temporal windows, as their analysis likely included both mixed Rossby-gravity and QBWO signals.





## 4 Moisture budget

As mentioned in the introduction, similar to MJO/BSISO, high coherence between moisture and convection anomalies has been observed for the QBWO in some ocean basins. We start by checking whether this is also applicable for the QBWO over the SWIO. Comparing specific humidity anomaly composites in Figure 4 with the OLR anomaly composite in Figure 2, it is clear that filtered OLR is highly co-located with filtered column-integrated specific humidity. Thus, the QBWO in this basin is a likely candidate for the application of moisture mode framework. In essence, an understanding of how moisture evolves should help in capturing the evolution of the convection. We now present a detailed moisture budget analysis of this mode. Specifically, when we have a large negative OLR anomaly associated with strong convection, it is accompanied by large positive anomalous column-integrated specific humidity and vice-versa. So, it is appropriate to perform a column-integrated moisture budget of the QBWO, and the relevant equation reads,

$$[\frac{\partial q'}{\partial t}] = -[(\mathbf{V}.\nabla_h q)]' - [(\omega \frac{\partial q}{\partial p})]' - P' + E' + R, \tag{1}$$

where $q$ is the specific humidity, $\mathbf{V} = u\mathbf{i} + v\mathbf{j}$ is the horizontal wind, $\nabla_h = \mathbf{i}(\frac{\partial}{\partial x}) + \mathbf{j}(\frac{\partial}{\partial y})$ is the horizontal gradient operator, $P$ is precipitation, $E$ is evaporation, and $\omega$ is the vertical velocity in pressure co-ordinates. Here, prime denotes a 10-25 day (QBWO) anomaly. $R$ is the residual in the budget. The square bracket represents mass-weighted vertical integrals, calculated from 1000 to 200 hPa. The last three terms of the R.H.S are usually bundled together as,

$$-[Q_2]'/L = -P' + E' + R, \tag{2}$$

which is often called the column-integrated "apparent moisture sink" (Adames and Wallace, 2015). The last four terms in Equation 1 are together called "column-processes" and this can be expressed as,

$$C' = -[(\omega \frac{\partial q}{\partial p})]' - P' + E' + R = -[(\omega \frac{\partial q}{\partial p})]' - [Q_2]'/L. \tag{3}$$

Hence, this term can be calculated directly by subtracting horizontal advection from moisture tendency. As precipitation and evaporation are not defined at pressure levels, the moisture budget equation at a single pressure level is often written as,

$$\frac{\partial q'}{\partial t} = -(\mathbf{V}.\nabla_h q)' - (\omega \frac{\partial q}{\partial p})' - Q_2'/L, \tag{4}$$

which we have used for our vertical structure investigation.

From Figure 4 one can see that sudden moistening between 10-20S and around 60E (mostly to the east of it) happened from Day $-4$ to Day $-1$, which caused the first southward excursion of the convection. To understand the cause of this moistening, we present the moisture budget for Day $-3$ in Figure 5. This figure shows that horizontal moisture advection (Figure 5b) is the primary cause behind moistening toward the eastern side of Madagascar, though 'column-process' (Figure 5c) also has a minor role. As expected, vertical advection (Figure 5f) and precipitation (Figure 5e) largely cancel each other, but overall, vertical advection is stronger, and along with a small contribution from evaporation, it causes some moistening through the 'column-process' term. On this day, we also see some moistening in the Mozambique channel, particularly to the north-west





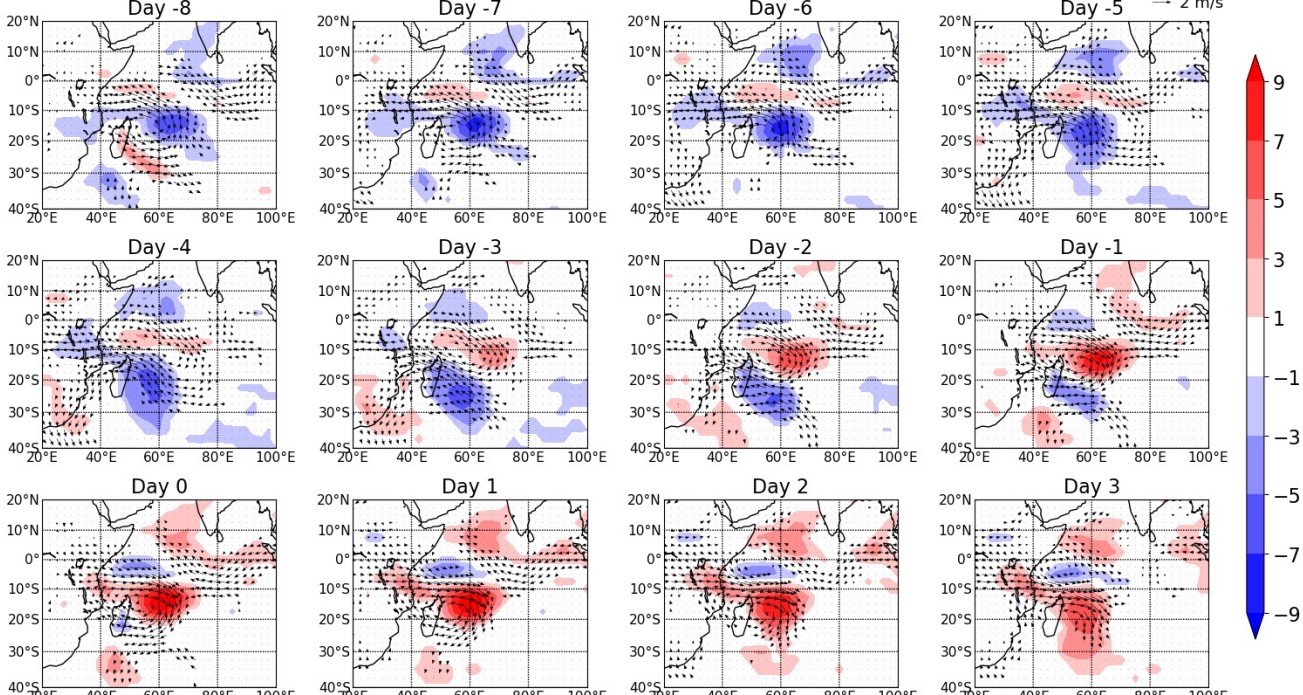

**Figure 4.** Composite of 10-25 day filtered column-integrated specific humidity (scaled by the latent heat of vaporization, $L$) ($10^6$J m$^{-2}$; shading) and 850 hPa wind anomalies (quivers). Wind vectors shown are statistically significant at 95% confidence level.

of Madagascar. This moistening is also dominated by horizontal moisture advection. We confirm as the vortex moves slowly southwestward, same processes continues for the next couple of days (not shown), with little more contribution from the 'column-processes', probably due to stronger convergence inside the gyre, which helps the convection (moisture) extend into the gyre from the eastern side (Figure 4). Going back to Figure 4, we see that the strengthening of the moisture anomaly continues on the eastern side of Madagascar up to Day 1, but we also see slight movement of moisture anomalies to the west. From Day 1 to Day 3, the moisture anomaly jumps beyond 20S and reaches 30S. To understand the moistening at this stage, we focus on the moisture budget of Day 2 (Figure 6). This again suggests that moistening is primarily due to horizontal advection. Now, to understand the specific processes behind this horizontal advection, we split this term into components comprising of QBWO and low-frequency background scales, but before that, to determine the vertical level where the moistening is most relevant, we briefly discuss the vertical structure of moisture budget along with the vorticity structure.

We have averaged over 55-65E to paint an average picture on Day $-3$ — as seen in Figure 7a, the moisture anomaly has mostly upright vertical structure, and so does the tendency (Figure 7b). Naturally, tendency is situated south of the highest moisture anomaly signal. Horizontal advection (Figure 7d) explains most of the tendency, while there is also some contribution from the column processes. Interestingly, highest moisture, moisture tendency, as well as horizontal advection are not in the



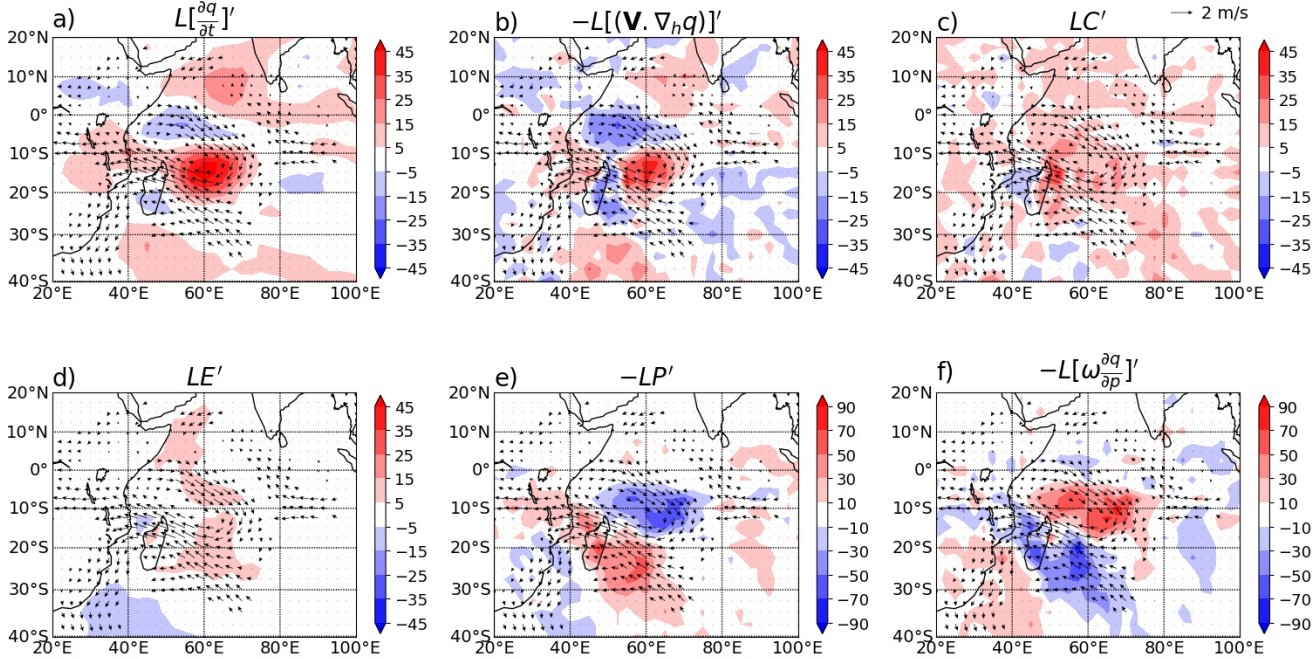

**Figure 5.** The composite mean of 10-25 day anomalous terms of the column-integrated moisture budget as in Equation 1 (scaled by the latent heat of vaporization) and their combinations (for column-processes) on Day -3, namely, (a) Moisture tendency, (b) Horizontal advection,(c) Column-processes, (d) Evaporation, (e) Precipitation, (f) Vertical advection. Units of terms are W m$^{-2}$. The 700 hPa wind anomalies are overlaid for reference. Wind vectors shown are statistically significant at 95% confidence level.

boundary layer, but just above it, in lower-to-middle free troposphere, which is line with the BSISO (Ghatak and Sukhatme,

2024). Thus, a choice of 700 hPa for analysis as representative level makes sense. Vorticity signal (Figure 7e) associated with the cyclonic circulation shows barotropic to equivalent barotropic structure, as the sign changes above 300. Looking at the anomalous moisture and vorticity structures, we can observe as the vortices gets more and more coupled with moisture (away from the equator in this case), it becomes more baroclinic, which is inline with the previous observations of many of the CCER waves/QBWO (Yang et al., 2007; Wheeler et al., 2000; Chen and Sui, 2010). Anomalous vertical velocity (Figure 7f) shows

an upright structure, and it is mostly in phase with the moisture anomalies (negative pressure velocity means upward motion).

## 4.1   Process of moistening

From the moisture budget analyses, it is clear that the horizontal advection is the main contributor to moistening that drives the south-southwestward propagation of convection. To identify the specific processes responsible for the anomalous horizontal



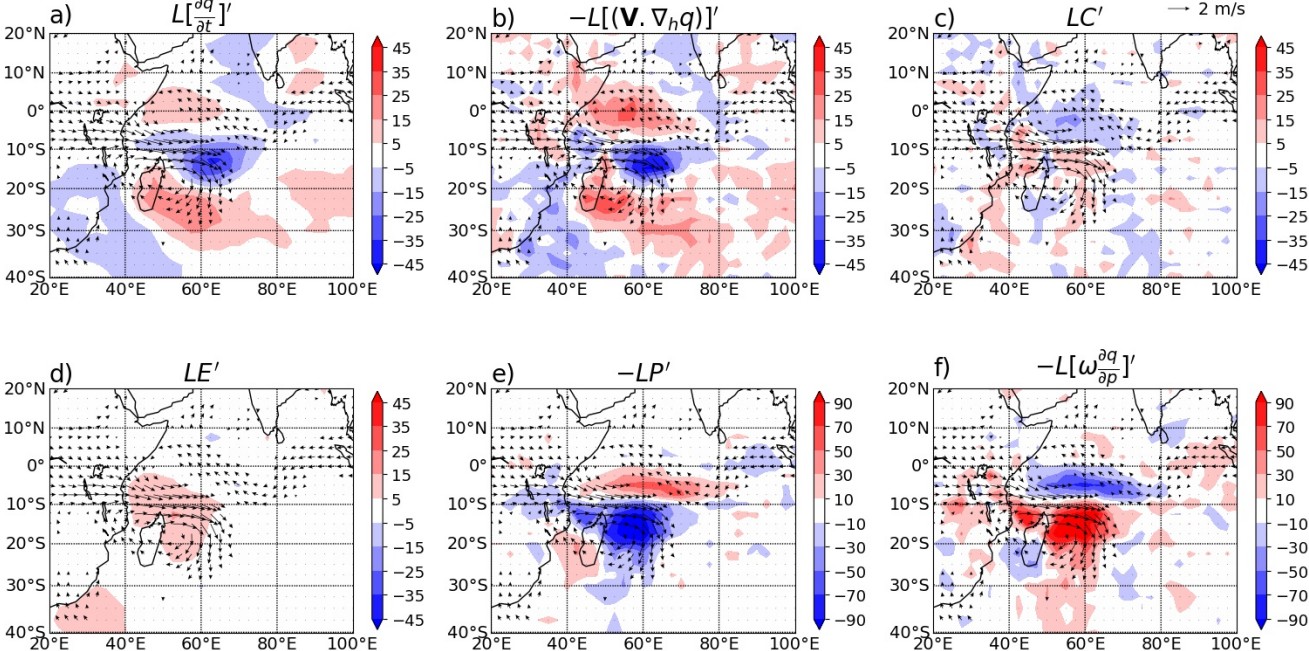

**Figure 6.** Same as Figure 5, but for Day 2.

advection we decompose it into terms consisting of QBWO-scale and background state wind and moisture components, i.e.,

$$(\mathbf{V}.\nabla_h q)' \approx (\mathbf{V}'.\nabla_h \bar{q})' + (\bar{\mathbf{V}}.\nabla_h q')' + (\mathbf{V}'.\nabla_h q')', \tag{5}$$

where prime means the QBWO-scale perturbation (10-25 day filtered anomaly), and bar refers to the slowly varying seasonal background (mean plus first three harmonics). Of course, there are contributions from other timescales, but they are smaller than the terms shown, so Equation (5) is a good first-order approximation. In fact, the last term on the R.H.S. is much smaller than the first two terms, and we have observed that these two terms together capture most of the QBWO horizontal moisture advection anomaly. Physically, the first term in the R.H.S. is the background moisture advection by the anomalous QBWO wind, and the second term is the anomalous QBWO moisture advection by the background wind. We focus on one level (namely 700 hPa), as this is the level where both moisture tendency and moisture advection are large as seen in Figure 7, so we can use this level as the representative level in context of moist processes. We have chosen the same days we used in our moisture budget analysis to understand the importance of the two processes in different stages of propagation.

Examining Figure 8, one can see that the summation of two decomposed terms (Figure 8d) more or less capture the pattern of the full horizontal advection term (Figure 8a), though the value of the full positive moisture advection to the eastern side of the Madagascar (the red patch, particularly to the east of 60E) is stronger than the summation of decomposed terms. This indicates, on Day −3, though the decomposition can explain the horizontal moisture advection pattern to the first order, there are also



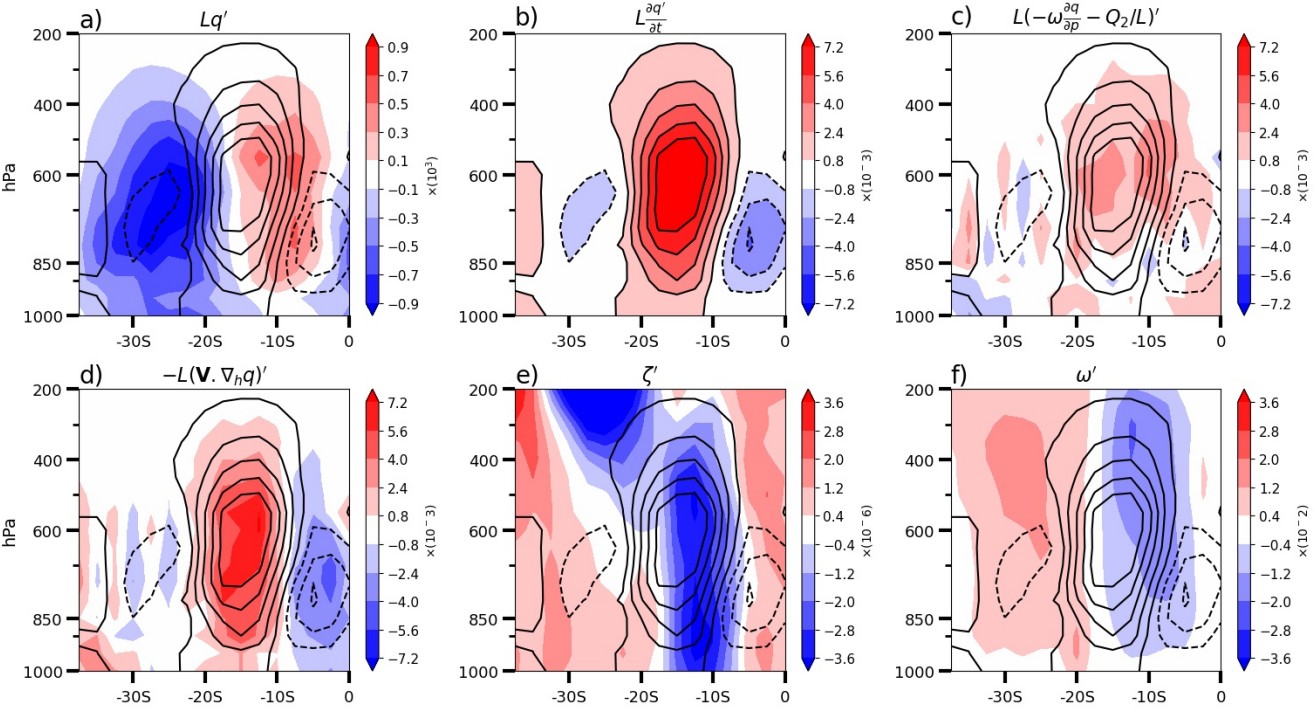

**Figure 7.** Pressure-latitude profiles of 10-25 days anomalous moisture budget terms in Equation 4 (scaled by latent heat of vaporization) and anomalies of a few other important variables averaged between target longitudes (55°-65°E) on Day -3 of the composite. The terms are namely (a) Specific humidity, (b) Moisture tendency, (c) The sum of vertical moisture advection and apparent moisture sink, (d) Horizontal moisture advection, (e) Vorticity, (f) Vertical velocity in pressure coordinates. Moisture tendency contours are overlaid on moisture budget terms. Units for specific humidity (moisture) is J kg$^{-1}$, vorticity is s$^{-1}$, vertical velocity is Pa s$^{-1}$, and for all other terms is W kg$^{-1}$.

small contributions from other scale-interactions. It is clear that the advective moistening to the east of 60E primarily happens

due to the QBWO winds (primarily meridional wind) advecting background moisture (Figure 8b), while significant contribution comes from the background wind advecting QBWO moisture (Figure 8c) to the western side of the zone of advection by QBWO northerlies. This is different from studies on CCER waves/QBWO in different locations, where it is found that advection of anomalous moisture by the background wind is the primary contributor behind the propagation of convection (Mayta et al., 2022; Mayta and Adames Corraliza, 2024; Dong et al., 2024). Rather, this result is somewhat inline with the recent study over

the same basin by Yang et al. (2023). Proceeding to Day 2 (Figure 9), we again see that the background moisture advected by the QBWO winds is the dominant term that moistens the area east of Madagascar and south of 20S (Figure 9b). Whereas, the QBWO moisture advection by the background wind term provides moistening west of 60E (Figure 9c), hugging the coast of Madagascar, but this acts against the moistening to the east of 60E, thus limiting the moistening to some extent in that region.

To illustrate how the QBWO wind anomalies (i.e., the eddies) advect background moisture and background winds advect

QBWO moisture anomalies in their respective places of dominance, we plot 700 hPa background moisture along with 700





hPa QBWO wind anomalies as well as 700 hPa QBWO moisture anomaly with background wind on Day $-3$ and Day 2 for the composite in Figure 10. On Day $-3$, the tilted QBWO cyclonic gyre is situated on the north-western side of Madagascar. Given the orientation of background moisture, the QBWO northerlies to the eastern side of the vortex coincide with strong meridional gradient of the background moisture (Figure 10a), thus they advect the background moisture southward w.r.t the zone of strongest background moisture between 10-20S, and rapid moistening and strengthening of convection is observed in that region. On the other hand, there is a background anticyclonic circulation between 15-40S, with easterlies along its northern side (Figure 10b). Given the location of the negative moisture anomaly adjacent to the eastern coast of Madagascar, and nascent positive moisture anomaly to the eastern side of 60E, these easterlies result in positive moisture advection between 10-20S, particularly to the west of the nascent convection, which to some extent helps convection move westward (Figure 10b). We confirm that as the vortex slowly traverses southwestward, similar processes continue for next two days, which helps convection to move south-westward compared to its position on Day $-4$ or Day $-3$. Particularly, once the convective anomaly starts growing, the advection by the background wind becomes stronger, and as the convective (positive moisture) anomaly is preferentially located on the eastern side of the vortex, the background winds advect QBWO moisture anomaly into the vortex. Along with moistening from the 'column-processes' the area of convection extends towards the centre of the vortex in this phase of the QBWO, though eastern side of the vortex is still remains the zone of strongest convection (Figure 4).

On Day 2, the cyclonic circulation has moved further south-westward, and now the northerlies to the eastern side of the QBWO vortex are situated on the eastern side of Madagascar and extend almost up to 30S (Figure 10c). Again, these northerlies act on the meridional gradient of the background moisture, and moisten the region. The QBWO moisture anomalies now have shifted south-westward, to the eastern side of the Madagascar, and the background easterlies result in moistening adjacent to the eastern coast of Madagascar (Figure 10d). Thus, the moisture tendency in the QBWO over the SWIO is primarily due to horizontal advection in the lower free troposphere, and both eddy advection of background moisture and the background winds advecting anomalous moisture are important contributors to the moistening process.

## 5   Vorticity budget of the QBWO

We now analyse the propagation of the QBWO vortex by constructing a budget of the vorticity anomalies of the system. The relevant equation reads (Wang and Chen, 2017),

$$\langle\frac{\partial\zeta}{\partial t}\rangle' = \langle(-\omega\frac{\partial\zeta}{\partial p})\rangle' + \langle(-\mathbf{V}.\nabla_h\zeta)\rangle' + \langle(-v\frac{\partial f}{\partial y})\rangle' + \langle-(\zeta+f)D]\rangle' + \langle T\rangle' + \text{residual}, \tag{6}$$

where, $\zeta = (\frac{\partial v}{\partial x} - \frac{\partial u}{\partial y})$ and $D = (\frac{\partial u}{\partial x} + \frac{\partial v}{\partial y})$ are the relative vorticity and divergence, respectively. $\mathbf{V} = u\mathbf{i} + v\mathbf{j}$ is the horizontal wind, $\nabla_h = \mathbf{i}(\frac{\partial}{\partial x}) + \mathbf{j}(\frac{\partial}{\partial y})$ is the horizontal gradient operator, $f$ is Coriolis parameter and $\omega$ is the vertical velocity in pressure co-ordinates. Prime denotes a 10-25 day anomaly as defined earlier. $T$ is given by $[(\frac{\partial\omega}{\partial y})(\frac{\partial u}{\partial p}) - (\frac{\partial\omega}{\partial x})(\frac{\partial v}{\partial p})]$, which is the tilting term, $[-(\zeta+f)D]$ represents the stretching term, $\frac{\partial\zeta}{\partial t}$ is the local tendency of the relative vorticity, $(-\mathbf{V}.\nabla_h\zeta)$ and $(-\omega\frac{\partial\zeta}{\partial p})$ represent the horizontal and vertical advection of relative vorticity, respectively, and $(-v\frac{\partial f}{\partial y})$ is the vorticity generation due to




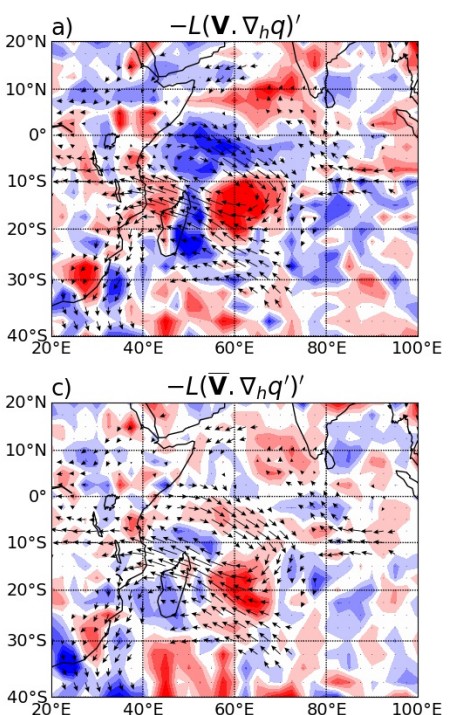
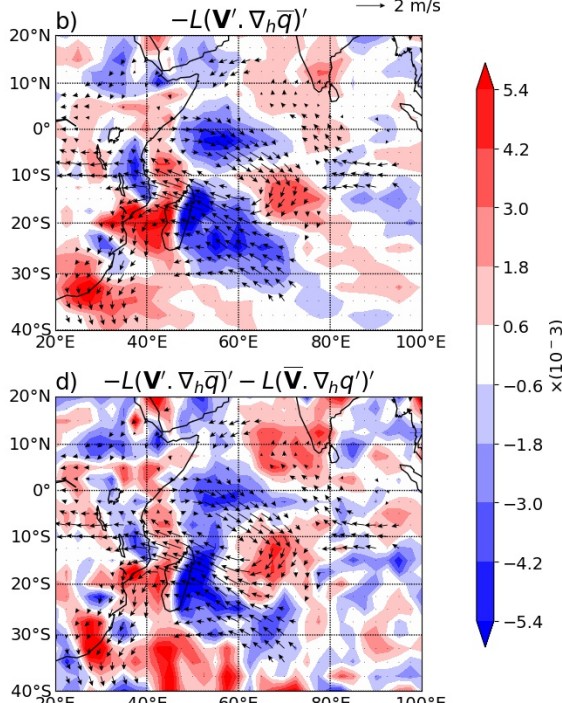

**Figure 8.** (a) 10-25 day anomalous horizontal moisture advection term and its decomposed primary contributor terms, namely (b) Background moisture advection by QBWO anomalous winds, (c) QBWO anomalous moisture advection by background winds and (d) their combination (all scaled by L) at 700 hPa as shown in Equation 5, for Day -3 of the composite. Units of terms are W kg$^{-1}$. The 700 hPa wind anomalies are overlaid for reference. Wind vectors shown are statistically significant at 95% confidence level.

the $\beta$ effect. Finally, residual denotes the unresolved portion of the vorticity budget in the re-analysis data. Angle bracket here represents mass-weighted vertical integral in the lower free troposphere (850 to 500 hPa). We have chosen this levels as these are the levels where anomalous moistening and moisture are strongest, and the vorticity structure is more or less homogeneous
(Figure 7).

In the previous section, we discussed how the northerlies of the eastern part of the cyclonic vortex (to the eastern side of 60E) act on the background moisture gradient to advect moisture and cause the maturation of convection between 10-20S. It is important to note, that on Day $-6$ or Day $-5$, when the vortex first became clearly visible, it was situated to the northeast relative to its position on Day $-3$ (Figure 4). Thus, this southwestward movement of the cyclonic vortex brought the northerlies
to the location of strong meridional gradient of background moisture , which caused the moisture advection to give rise to the convection in the location where the system matured. Naturally, we ask the question, what causes this southwestward movement of the vortex, which is essentially responsible for the strengthening of convection in that particular region.



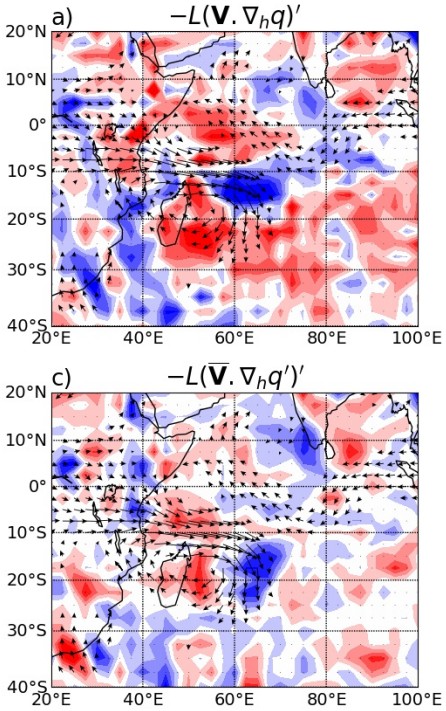

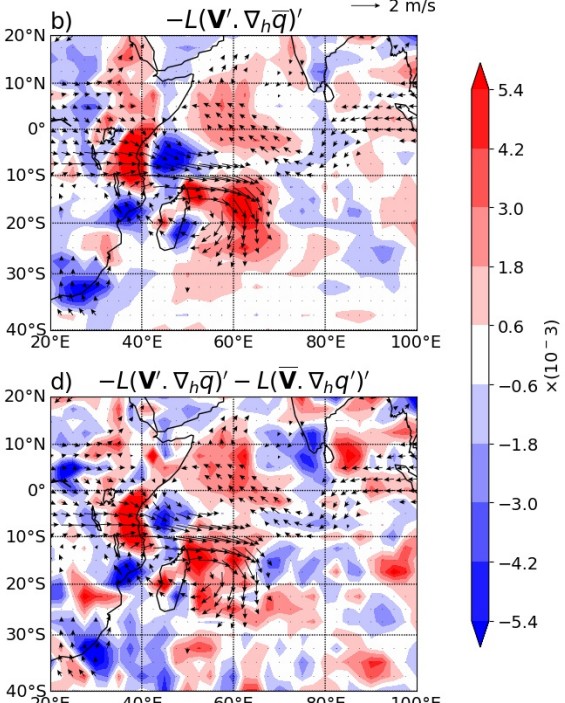

**Figure 9.** Same as Figure 8, but for Day 2.

To answer this question, we appeal to a vorticity budget of the QBWO. Figure 11 shows the terms comprising the lower free-tropospheric vorticity budget and their combinations, along with vorticity itself on Day $-5$. Here, we see that the tendency of relative vorticity (Figure 11b) has a minimum to the southwest of the tilted cyclonic vortex (Figure 11a), and is largest northeast of the vortex centre. This is consistent with the fact that the cyclonic vortex is moving towards the southwest. The only term that matches this pattern, i.e., has a dipole structure with a minimum in the southwest and maximum in the northeast is the $\beta$ effect term (Figure 11c), though its spatial extent is much larger than the vorticity tendency. The form of the $\beta$ term results from the meridional velocity associated with tilted wavetrain comprising the cyclonic gyre to the northeast, and the anticyclonic gyre to the southwest. The horizontal advection term (Figure 11f) is also large northeast of Madagascar, where the negative vorticity tendency is largest. The cause for this pattern is also understandable as the background easterlies blow from a lower vorticity region to higher vorticity region in this sector (see Figure 10d and 11a). Thus these two terms together help the vortex to move southwest; but when we add these two terms, the resulting combination yields much stronger and spatially larger pattern than the tendency (Figure 11g). We note that the stretching term (Figure 11d), however, has a maximum almost in the same region as the minimum of the combination of the $\beta$ and horizontal advection terms, and it has a weak minimum to the northeast of it. This large positive stretching is likely a direct effect of the divergence associated with the positive OLR (negative moisture) anomaly situated in the same location (Figure 2 and Figure 4). Thus, stretching counteracts the combination of the $\beta$ and



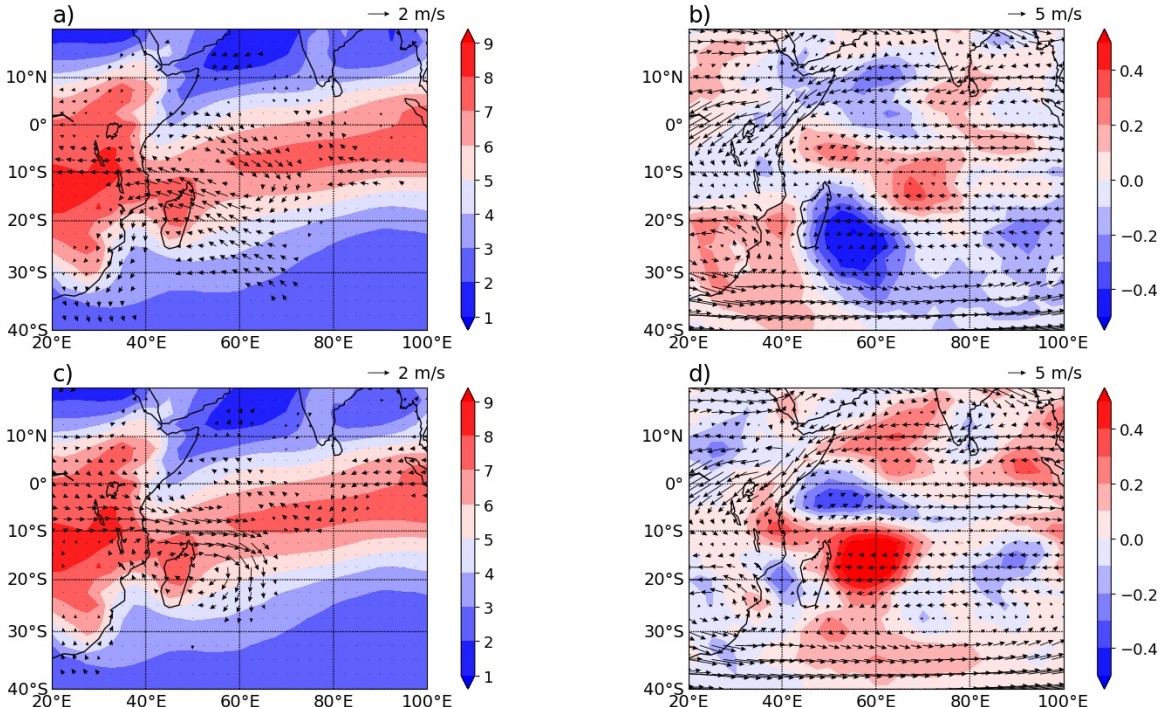

**Figure 10.** (a) Background specific humidity (g kg$^{-1}$) and 10-25 day filtered wind anomalies, (b) Background winds and 10-25 day filtered moisture anomaly for Day $-3$ of the composite at 700 hPa. (c) and (d) are same as (a) and (b) respectively, but for Day 2. Wind vectors shown are statistically significant at 95% confidence level.

horizontal advection terms, and limits their influence to the northeast of Madagascar. To a large extent, the combination of these three terms (Figure 11e) captures the vorticity tendency, though, as expected, it is more patchy than the tendency.

An interesting feature to note here is, compared to near the equator (Figure 11b), where the $\beta$ term largely shapes the tendency, the tilt of the negative tendency (blue region) is weaker, i.e., it is more zonally oriented. This happens because stretching limits the southward extent of the $\beta$ term. This reduction of tilt in tendency is reflected in the nature of the vortex over the few next days; specifically, compared to Day $-5$, the vortex is much more zonally oriented on Day 0 (Figure 4). Essentially, vortex stretching associated with the convective anomaly slows down the southwestward propagation of the QBWO.

To understand the interaction of moisture and circulation in the context of CCER waves, a moisture-vortex instability (MVI) mechanism has been recently proposed (Mayta and Adames Corraliza, 2024). Though originally discussed in the context of zonal propagation, their argument can be applied here too, with some adjustment due to the poleward component of propagation. In MVI, mean moisture is assumed to be highest at the equator and reduces with latitude, thus the moisture gradient is purely meridional. Now, zonally symmetric ER waves have southerly (northerly) winds towards the east of the cyclonic vortex

in the Northern (Southern) hemisphere, which moisten the area and cause convection after a time lag necessary for the moisture




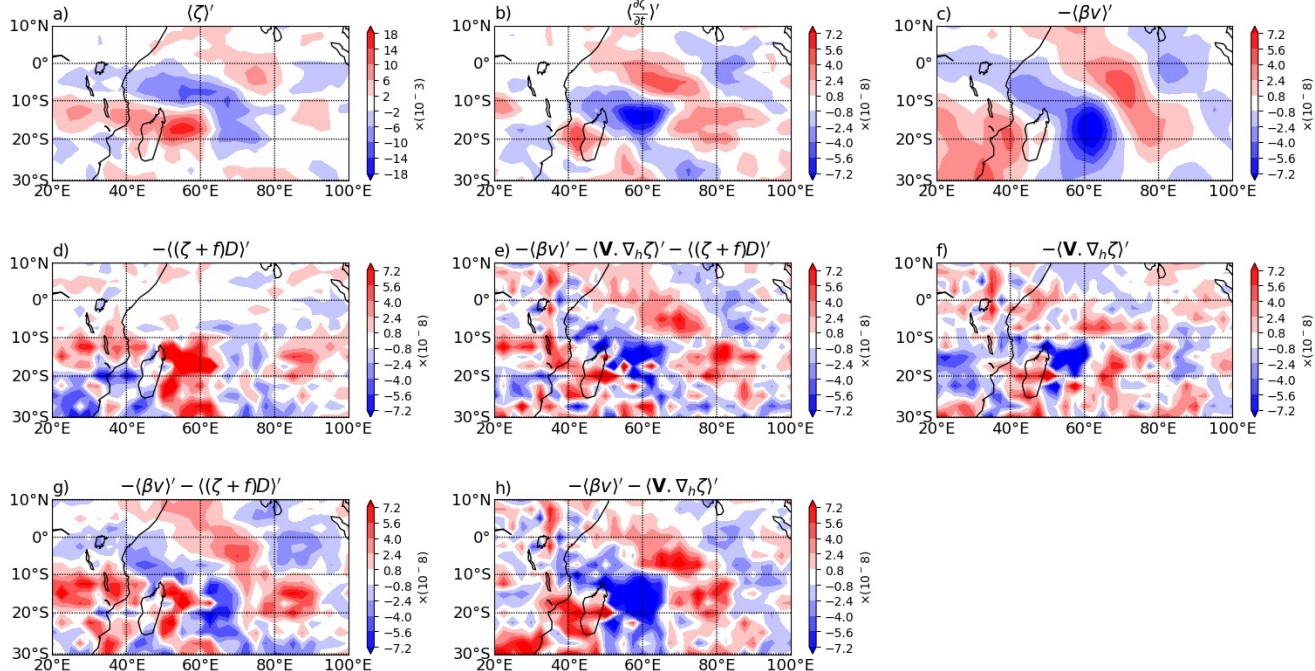

**Figure 11.** 500-850 hPa (lower free-troposphere) integrated 10-25 day (a) Vorticity anomaly and dominant vorticity budget terms for boreal winter(DJFM) as shown in Equation 6 and their combinations, namely, (b) Vorticity tendency, (c) $\beta$ term, (d) Vortex stretching, (e) Sum of $\beta$ term, vortex stretching and horizontal advection, (f) Horizontal advection, (g) Sum of $\beta$ term and stretching term, (h) Sum of $\beta$ term and horizontal advection, Day $-5$ of the composite. The unit of column integrated vorticity is $\mathrm{kgm}^{-2}\,\mathrm{s}^{-1}$ and of budget terms are $\mathrm{kgm}^{-2}\mathrm{s}^{-2}$.

buildup. According to this theory, stretching caused by this convection would propel the vortex towards the east in absence of any background flow, and due to some lag between moistening and convection, though convection will develop in the eastern side of the vortex, it will occur slightly inside the gyre closer to the vortex centre. Thus, in this setup, if the convectively coupled vortices move westward, it is due to a strong easterly background flow.

As horizontal moisture advection, vorticity advection by the background wind and convection generated stretching, all ingredients of this theory are present and important in our case, we examine if this argument holds here using the vorticity budget. Particularly, we focus on the movement of the cyclonic vortex. In the budget of Figure 11 stretching associated with above-average convection is weak as the negative OLR anomaly is small on Day $-5$ (Figure 2), i.e., it doesn't generate strong stretching to the east of the cyclonic vortex. Thus, we choose Day $-2$, when convection becomes stronger and the vortex is

convectively coupled (Figure 2). The story is more or less similar to Day $-5$, the $\beta$ term (Figure 12c) contributes to the negative tendency (Figure 12b) southwest of the cyclonic vortex, horizontal advection also has a role, though smaller than Day $-5$, and finally stretching makes a small contribution to the budget. Again, stretching (Figure 12d) associated with the positive OLR anomaly counteracts $\beta$ term and horizontal advection southeast of Madagascar.





To study the validity of the MVI mechanism discussed above, we focus on Figure 12g, where we add the stretching and $\beta$ terms,
thus this shows the effect of moist convection in isolation from the background wind. While the stretching (negative magnitude)
caused by the moist convection (Figure 2 and Figure 5) is situated towards the east of the cyclonic vortex (Figure12d), as
suggested by the MVI argument; it can't counteract the positive $\beta$ term (Figure12c) to cause eastward movement (with respect
to the background wind), thus the $\beta$ term wins over the stretching to the east-northeast of the cyclonic vortex and causes
a net positive value to the east-northeast and negative value to the southwest. This suggests that even without the effect of
the background wind, the vortex would move southwest, though much slower than a dry wave. In effect, stretching can't
overrule the effect of $\beta$ term, but opposes it, and slows down the system. This indicates one possible reason why the CCERs
are slower than their dry counterparts (even in the absence of any background flow), other than the equivalent depth argument
offered by the traditional (non-moisture mode) theories. Note that, horizontal advection (Figure12f) helps in the south-westward
movement, and it is more or less in-phase with the tendency, but it is not the primary reason for the southwestward movement
as indicated by Mayta and Adames Corraliza (2024).

To showcase another example of the interaction of $\beta$ and stretching, we also present the vorticity budget on Day 0 (Figure 13).
Here, we see the QBWO cyclonic vortex over the northeast of Madagascar (Figure 13a), and as before, the negative tendency
is situated southwest of the cyclonic vortex, over the island of Madagascar and Mozambique channel. The $\beta$ term explains this
negative tendency, stretching and horizontal advection don't have much influence on it at this stage. This is understandable as
the positive OLR anomaly has weakened over this region (Figure 2), stretching is weak, and over Madagascar and Mozambique
channel, the background wind is also very small (Figure 10b), causing weak horizontal advection. More carefully, we note that
the $\beta$ term is strongly negative where negative tendency is largest, as it is the region of the south-easterlies associated with the
cyclonic vortex (Figure 2). Here too, stretching (Figure 13d) is strongly negative east of the cyclonic vortex, associated with
strong convective anomalies, but the $\beta$ term is strong enough to overwhelm the stretching term, and the horizontal advection
also opposes stretching. In all, in the presence of moist convection, stretching opposes $\beta$ as in MVI, but it does not overcome
the tendency induced by the planetary vorticity.

## 6 Discussion and Conclusions

In this study, we have identified a southwestward propagating convectively coupled quasi-biweekly oscillation (QBWO) over
the South-West Indian Ocean (SWIO) during boreal winter. A composite drawn from 30 years of data shows that the QBWO
exhibits distinct propagation characteristics in filtered OLR and low-level circulation. On an average, the QBWO gyres have
a northwest-southeast tilt and they move southwestward with a westward phase speed of about 3.5 m s$^{-1}$. The system has a
time-period of about 14-18 days and a zonal planetary wavenumber 5-6 with negative OLR anomaly lobes as well as associated
circulation gyres spanning approximately 30-35 degrees in longitude and 20 degrees in latitude, respectively.





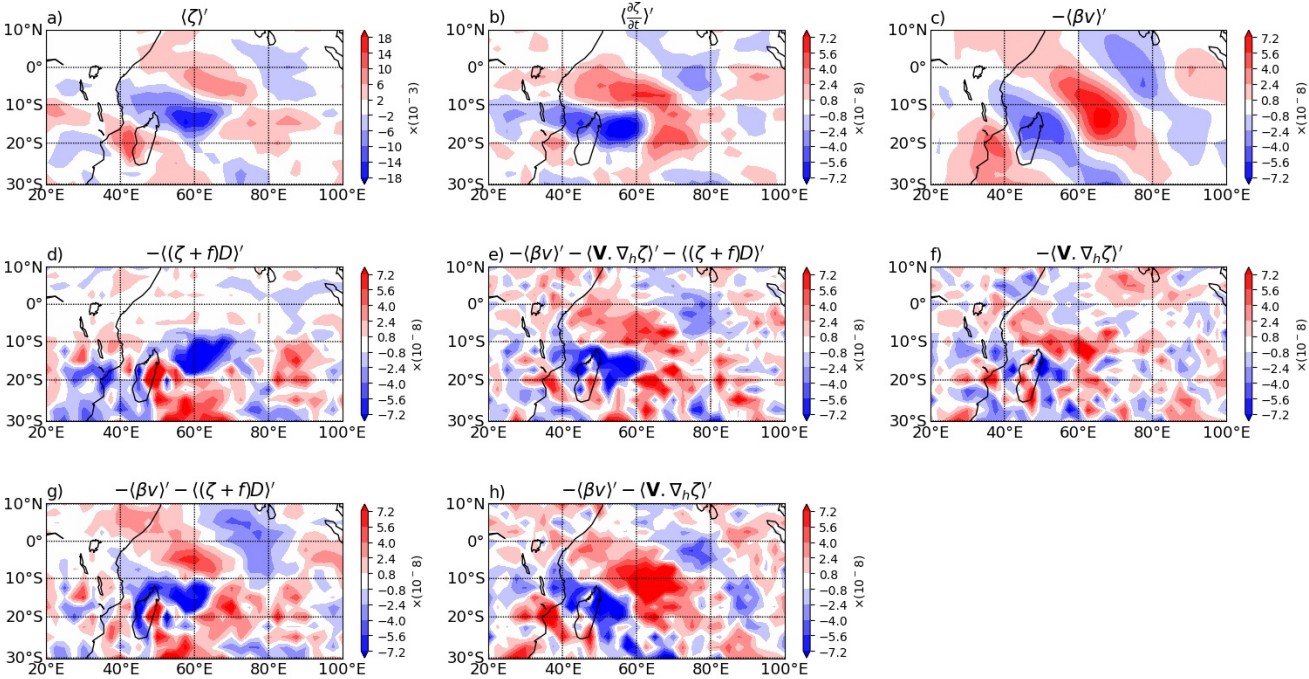

**Figure 12.** Same as Figure 11, but for Day $-2$

The wavenumber, phase speed, and time-period of this QBWO in the SWIO fall within the range reported for convectively

coupled gravest ER waves (Wheeler et al., 2000; Kiladis et al., 2009; Janicot et al., 2010; Mayta et al., 2022; Molinari et al., 2007) and QBWOs documented in various geographical regions (Chatterjee and Goswami, 2004; Kikuchi and Wang, 2009; Wang and Chen, 2017). The baroclinic nature of the QBWO when strongly coupled with convection, and near-barotropic structure while loosely coupled/uncoupled with convection has also been observed in CCER waves (Wheeler et al., 2000; Yang et al., 2007; Kiladis et al., 2009). Here, during the boreal winter, the QBWO is restricted to the Southern Hemisphere, and

its propagation has a prominent poleward component along with westward movement. Though idealized dry ER waves are symmetric about the equator and move purely westward (Matsuno, 1966), and studies have also observed almost symmetric westward propagating convectively coupled ER waves (Nakamura and Takayabu, 2022), this kind of asymmetric wave structure of CCER waves which is dominated by one hemispheric component and propagates poleward along with westward movement is anticipated and documented in different background conditions (Xie and Wang, 1996; Zhang and Webster, 1989; Wheeler

et al., 2000; Molinari et al., 2007; Mayta et al., 2022). Indeed, similar structures of the QBWOs has been reported in various basins in the northern hemisphere (Chen and Sui, 2010; Wang and Chen, 2017). The ER wave in the SWIO documented by Bessafi and Wheeler (2006) is also quite similar to our results, including the position of strongest convection, except the prominent poleward propagating component, probably because of the symmetry condition imposed while isolating the waves. Taken together, the structure, features and dynamics of the QBWO over the SWIO in the boreal winter has many similarities





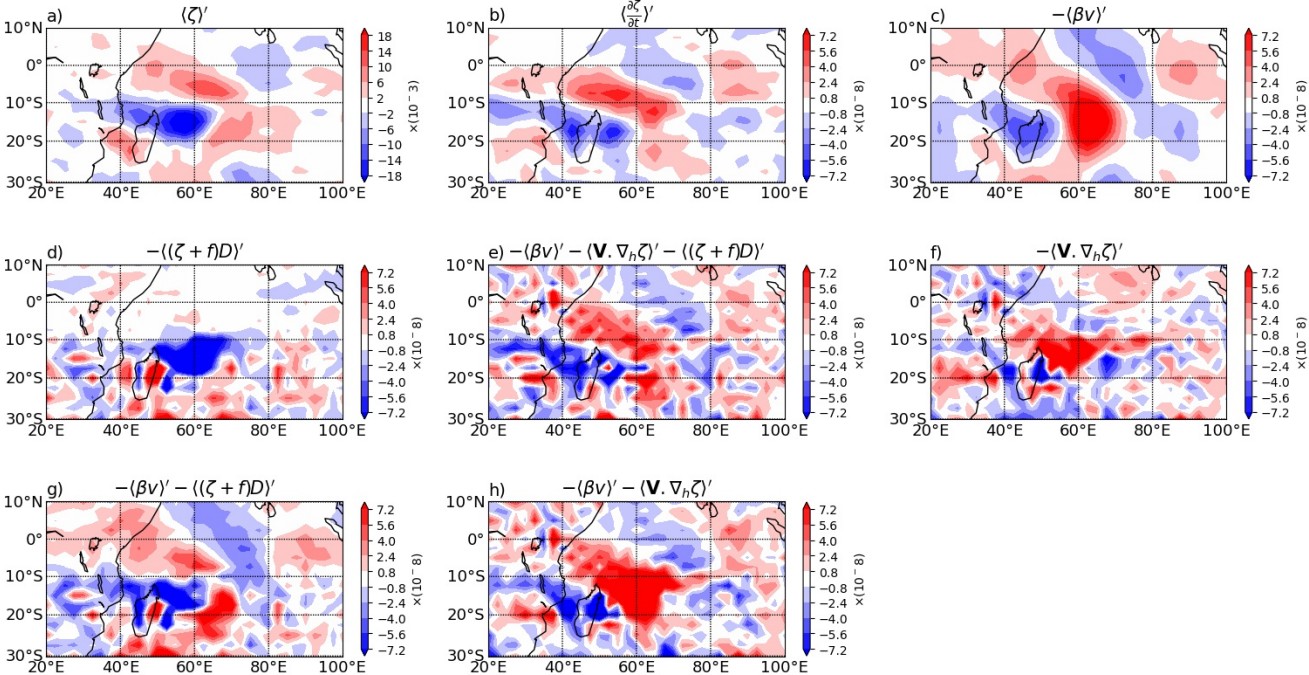

**Figure 13.** Same as Figure 12, but for Day 0

with its Northern Hemisphere counterparts, and appears to be consistent with many convectively coupled ER wave properties, modified by existing background states.

There are quite a few important properties of observed CCER waves & the QBWOs which deviate prominently from dry ER waves (Matsuno, 1966), and are yet to be clearly understood. These include the observed poleward propagation in a few basins, the slow phase speed of the system and phase relation between circulation and convection. Traditional views tried to understand the slower phase speed by reduction of the equivalent depth, but that is not always consistent with observations (Mayta and Adames Corraliza, 2024). Moreover, the cause of the observed phase relation between circulation and convection remains unclear (Chen, 2022). In this context, the moisture mode framework has recently shown promise to understand slowly evolving tropical systems; where moisture is prognostic, and coupling between moisture and circulation dictates the properties of the moist wave. Observing a high coherence between anomalous moisture and convection, we have used this framework to understand the documented QBWO in the SWIO.

Our analysis reveals that the QBWO vortex starts with a NW-SE tilt near the equator, which, in a way, has an influence on its propagation. When the tilted cyclonic vortex is close to the equator, it has very weak associated convection. Subsequently, the system moves southwestward due to the $\beta$ term (which arises because of the initial tilt of the vortex, as southerly winds of this cyclonic vortex as well as previous anticyclonic vortex is situated to the southwest of the cyclonic vortex), as well as



the horizontal advection of vorticity by the background wind. These two terms together overcome the large stretching term associated with the dry (anticyclonic) anomaly and propel the cyclonic vortex southwestward. Later in its life-cycle, when the cyclonic vortex centre reaches around 10S, 60E, anomalous northerlies near the eastern flank of the vortex tap the background meridional moisture gradient and QBWO convection starts strengthening, particularly to the eastern side of the vortex. This strengthening convection generates large negative stretching towards the eastern side of the vortex.

At this stage, the moisture vortex instability (MVI) view (Mayta and Adames Corraliza, 2024) predicts that this stretching should overcome the $\beta$ effect, and in absence of the background easterlies, the vortex would move eastward. However, here we find that stretching, though strong, is not enough to counter the $\beta$ effect, and even if we don't include horizontal advection, the resultant would generate vorticity tendency towards southwest, rather than southeast. Further, With the inclusion of horizontal advection, the southwestward vorticity tendency becomes larger and the vortex continues to move towards southwest. The

northerlies continue to advect background moisture and further strengthen the convection on the eastern flank of the vortex. Now, as the convection gets matured to the eastern side of the vortex, the background easterlies advect anomalous moisture inside the vortex, and along with a contribution from the column process, we see that moisture and convective anomalies extend inside the vortex, though the strongest convection is still observed to the east of the vortex. The southwestward propagation of the vortex continues up to approximately 30S to the east of Madagascar, and the northerlies of the eastern side of the

vortex continue to advect the background moisture southward, and the background wind continues to advect QBWO moisture anomalies westward. Finally, after reaching 30S, the convection weakens and the whole system dies down.

In general, given that the background moisture decreases poleward, stretching generated by moist convection acts against the $\beta$ effect term of a vortex. (Mayta and Adames Corraliza, 2024; Ahmed, 2021). In our case, the $\beta$ effect wins, but stretching effectively slows the westward propagation. Although the ingredients are the same, this differs slightly from the MVI mechanism

for CCER waves suggested by Mayta and Adames Corraliza (2024), where it was suggested that the background easterlies are necessary to explain the westward propagation; as stretching overrules the $\beta$ effect, it would cause eastward propagation in the absence of the background easterlies. However, as we have shown here, that there can be convectively coupled cases, where stretching due to moisture coupling doesn't overcome the $\beta$ effect term, but weakens its influence on the vorticity dynamics. In such cases, these two terms are sufficient to cause slow westward propagation, and horizontal advection by the background

winds can have an assisting role to increase the westward phase speed. In essence, our results can be seen as a minor extension under the umbrella of the MVI mechanism (Mayta and Adames Corraliza, 2024), which can be tested in different geographical locations.

Overall, the southwestward propagating QBWO over the SWIO during the boreal winter is a result of active interaction between prognostic moisture and vorticity dynamics. Indeed, this is an example where moisture and the circulation influence

each other to dictate the propagation of the convectively coupled system. It is worth noting that general circulation models still have difficulties in producing an accurate QBWO (Jia and Yang, 2013; Wang and Zhang, 2019). The understanding of basic mechanisms involved in the QBWO offered by this study, particularly in the SWIO basin may help diagnose the issues that plague these efforts, and will contribute to enhanced intraseasonal prediction capabilities, especially in the context of Mada-



gascar and parts of south-east Africa, where the boreal QBWO has significant influence on the regional climate, influencing
subseasonal rainfall variability as well as extremes such as tropical cyclones.

*Code and data availability.* All data used in this study are publicly available. NOAA OLR data can be accessed from https://psl.noaa.gov/
data/gridded/data.interp_OLR.html, ERA5 data can be accessed via https://cds.climate.copernicus.eu/cdsapp/dataset/reanalysis-era5-pressure-levels,
Details for accessing and using the Windspharm package Dawson (2016) are given at https://ajdawson.github.io/windspharm/latest/.

*Author contributions.* Both authors contributed equally to the work.

*Competing interests.* The authors declare no competing interests.

*Acknowledgements.* SG acknowledges financial assistance from Council for Scientific and Industrial Research (CSIR) and the Divecha
Centre for Climate Change, IISc. JS acknowledges support from the University Grants Commission, project F 6-3/2018 under the Indo-Israel
Joint Research Program ($4^{th}$ cycle). Both authors thank the reviewers for their constructive comments and their suggestion to examine the
moisture budget of the QBWO.



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
