# Peer review of "Southwestward propagating quasi-biweekly oscillations over the South-West Indian Ocean during boreal winter: Characteristics and propagation mechanism"

_EGUsphere, 2024_

## Author Comment (AC1)

**Final Response to Reviews**

(Please note that the Reviewer's comments are in normal font and our response is in italics)

**1 Response to Reviewer 1**

The authors do a comprehensive study on the dynamics of quasi-biweekly oscillations (QBWO) over the southern Indian Ocean. They find that moisture is central to their dynamics, and the beta effect critical to its propagation. The results here further support the notion that moisture is important to understand tropical dynamics, and I welcome this type of contribution. My main concern with the paper is that it is long and windy. The authors spend a significant amount of time describing the figures and the literature, often at the expense of getting to the point. I think the paper will benefit substantially from some thinning. Because of this I recommend major revisions. Below are my recommendations for the authors to consider.

*We thank the reviewer for encouraging comments on our work. We have addressed the concerns of the reviewer below.*

**Comments**:

1. Introduction is too long. With the exception of the last paragraph, the second half of the introduction (lines 90-124) do not add much to the context of the paper. The first few pagraphs motivate the paper, but then we go to talk about QBWO and its impacts. I think all these things can be summarized in a few sentences in the first two paragraphs, and once the discussion about how moisture mode theory may apply to QBWO then the authors can motivate the study and move to the next section. So much discussion here is not necessary.

*Thanks for the comment. We have shortened the introduction.*

2. Figure 2 contains 12 panels showing the daily evolution of the QBWO. Yet the system doesn't change all that much from day to day. Showing panels every other day can make this figure look less cluttered.

*We also thought of showing panels every other day, but we encountered a problem in that representation. Though at first glance, it looks like that the change is not rapid, but during some parts of the life-cycle, the daily changes are crucial, which complicates the explanation. For example, in Figure 4, if we focus on Day -4 and Day -2, winds of Day -4 can't explain the position of moisture on Day -2, for that, we need the QBWO circulation of Day -3, and the changes between Day -3 and Day -2 becomes important.*

3. From the discussion, it seems like you don't need to show both Figs. 5 and 6. Just focus on one of them. A summary figure like Fig. 9 of Mayta and Adames Corraliza (cited by the authors) can inform other days.

*We purposely avoided that kind of representation, as we have noticed this can be misleading. We have seen that in the case of ISOs, in many cases, the moistening process is quite different in different geographical locations. Condensing them might cause a loss of crucial information. In our case, there is no a priori way to know whether the processes are location specific or not. Though in our case, the moistening processes are not very different, they are not identical, and the importance of the advection by the background wind is more clear at Day 2. Also, showing these two plots helps us to bind the story well, thus we want to continue with both the figures. If the reviewer very strongly feels the need to delete one of the figures, we can shift it to the appendix.*

4. Figure 7 might need to be split. Panels a-d all show the moisture budget. However, in panels (e) and (f) it might be more useful to show the moisture anomalies rather than the tendency as it is more in line with what the authors discuss in the text.

*Sorry for the confusion here. We had another very important point that links the vorticity structure and the tendency, and directly links to our moisture budget analysis, but we missed the point in the manuscript. The point is, few studies on the QBWO, particularly for those which move poleward, sees it in the light of the theoretical work of Jiang et al.(2004), where boundary layer moisture convergence causes the moistening ahead of the existing convection, and the convergence is a result of generation of barotropic vorticity over the same location. Thus, one might expect same to be true to the south of the existing convection in our case. But here, though the negative vorticity slightly leads the positive moisture anomaly towards the south, clearly, between 15-20S, there is very strong moistening, but vorticity anomaly is very weak. Also, moistening is quite weak in the boundary layer compared to the free troposphere. Thus Jiang et al.(2004) style propagation mechanism is not applicable here. This is also evident from the upright structure of the vertical velocity and the moisture tendency, as, where boundary layer moisture convergence is prominent moistening process, we expect a tilted structure of moisture tendency and vertical velocity, as in the case of MJO. We have included this in the section 4 of the revised manuscript.*

5. 8 and 9: Same comment as figs. 5 and 6.

*Extension of the answer to comment 3. We need to show both the figs 5 and 6 if we keep figs 3 and 4.*

6. 11-13: Only need to show one. Can show a summary plot like I mentioned above for Figs 5 and 6.

*Same answer as the answer to comment 3. Still, to shorten the paper as indicated by the reviewer, we are shifting Figure 13 to the newly added appendix.*

**2 Response to Reviewer 2**

Based on the observation and reanalysis datasets, this study found a southwestern-propagating QBWO mode over the Southwest Indian Ocean characterized by the coupled convection and circulation anomalies, which have a typical period of 14-18 days and westward phase speed of approximately 3.5 m/s. A vorticity budget analysis reveals that the beta effect plays a leading role in southwestward propagation, while the moisture mode is essential in reducing the speed of propagation of the QBWO by acting against the beta effect of the vortex. The results are interesting and reliable. This paper is well-written and organized. Nevertheless, some improvements can still be made to improve the quality and clarity of the paper. I recommend a major revision for this round.

*We thank the reviewer for encouraging comments on our work. We have addressed the concerns of the reviewer below.*

**Comments**:

1. The author did not provide any power spectrum analysis but directly used a 10-25-day window to filter variables. If a more common 10-20 days filtering window is used, will it affect the results? If the authors want to validate the necessity of a 10-25-day window, it is recommended to add power spectrum analysis.

*We have described the rationale behind using a 10-25 day filter in Section 2. We have also mentioned that the result doesn't change significantly as long as we confine ourselves within 10-30 day band (lines 145 to 150 of the original manuscript), so that we don't interfere with the MJO or synoptic signals, and 10-30 day band of course includes 10-20 day filter that the reviewer mentioned. We confirm again, that the results don't change much if we use the 10-20 filter, still we prefer a little wider band of 10-25, for the reasons mentioned in Section 2. To validate, here we are presenting the power spectra of the OLR over the chosen box (Figure R1), and as we expected, there is considerable power around 24 days, so using a 10-25 filter makes sense. As indicated by the reviewer, we are adding this information in the section 2 (line 128) of the revised manuscript to make the validation point stronger, but we are not including the power spectra in the revised manuscript as Reviewer 1 wants us to shorten the paper to make it crisp.*

[Figure]

Figure R1: 30 years average power (W$^2$ m$^{-4}$) spectra of OLR averaged over the box chosen for the composite analysis during boreal winter (DJFM).

2. It is suggested that when analyzing moisture mode and vorticity dynamics, a profile section along the QBWO propagation path can be made, and then the evolution of different terms against time can be directly analyzed. This can avoid situations like Figures 5-7, which only present a one-day distribution.

*We had thought of this but deliberately avoided the suggested kind of representation, as the dominant vorticity and moisture dynamics can vary spatially (zonally and meridionally) in different geographical locations, thus analysis along a section can miss a lot of details, and it will be difficult to connect the effects of circulation and convection. For example, for a particular day, if two different terms are dominant at two different locations in the domain of interest (say one is dominant on the eastern side, and the other on the western side), but the section goes through a point which is dominated by only one term, the importance of the other term will be totally missed. Of course the budgets are shown for individual days, that is why, we show different budgets for different days (phases), so that our description doesn't get dominated by the dynamics of a particular phase. Keeping this in mind, we have continued with our present representation.*

3. Horizontal moisture advection (Eq. 5): Horizontal advection includes two components, zonal and meridional advection. Typically, zonal advection guides QBWO to propagate westward, while meridional advection leads QBWO to propagate northward/southward. The author should further decompose the specific roles of these two components. In addition, time scales lower than 10 days should also be included to explore some scale interactions with synoptic waves.

*Thank you. We agree with the reviewer's comment. From our plots, particularly from figures 8,9 and 10, it is intuitive that the mean moisture advection by QBWO wind is mostly dominated by the meridional component of the QBWO wind, as the background moisture gradient is mostly meridionally oriented (it is true for both the days shown), and the QBWO moisture advection by the background wind is dominated by the zonal component, as the background wind is mostly zonal in the region of interest. As the other reviewer wanted us to shorten the paper, we are not including this analyses in the main manuscript, but we have added it in the appendix (Figure A1 and A2). In the main manuscript, we have added a few comments on this issue at the end of section 4.*

*Regarding the scale interaction, we have found that there is some contribution from the scale interaction*

*between the synoptic (less than 10 days, denoted by double-prime) and the QBWO signals. Here (Figure R2), we have shown the result, and made one comment at section 4.1 around line number 300 in the revised manuscript.*

[Figure]

Figure R2: (Top left) 10-25 day anomalous horizontal moisture advection term, (Top right) QBWO anomalous moisture advection by synoptic (<10 days) winds, (Bottom left) synoptic(<10 days) moisture advection by QBWO anomalous winds, and (Bottom right) their combination (all scaled by L) at 700 hPa, for Day -3 of the composite. Units of terms are W kg$^{-1}$.

4. L352-L354: As shown in Figure 7e, the relative vorticity has a quasi-barotropic structure, so the vorticity integration from 1000hPa to at least 300hPa is suggested. Why did the authors choose the levels between 850hPa and 500hPa that have the strongest moisture signals to integrate vorticity? Is there a significant correlation or direct causal relationship between vorticity and moisture at these pressure levels?

*As discussed in the vorticity budget section of the preprint (see after line number 355), the basis of this analyses is the 'moisture mode' framework, thus we primarily wanted to explain the evolution of moisture. The moisture budget analyses leads us to the vorticity budget, as we have seen that the movement of the QBWO vortex causes the anomalous moisture movement (by advecting the background moisture) towards the south-west, thus we wanted to understand the reason behind the south-westward movement of the QBWO vortex. So, this vortex movement became a focus of our analyses. Moreover, moisture advection as well as anomalous moistening is strongest in the free-troposphere, as seen from Fig 7, thus it made sense to focus on the circulation of the free troposphere (500-850 hPa), as this is what caused the moistening. Though, as the reviewer pointed out, the vorticity doesn't change from 1000-300 hPa, thus integrating over these levels doesn't change the results in any significant manner.*

**Minor comments**

1. Figure 2: I suggest to add the propagation path indicators for convection and low-level circulation.

*Using two indicators would be confusing, as we are not claiming a direct and constant phase relation between these two. Rather, the phase relation and interaction between these two evolves throughout the lifetime, and both influence the other. Thus, we prefer to avoid using path indicator as that might be slightly misleading, and won't give any extra information, rather we feel it would complicate the description.*

2. Figure 3b: Only the positive V-wind anomalies have complete westward movement characteristics from 75E to 30E, while the negative ones before and after that are more like the standing waves and stop moving westward when reaching 50E. This may be attributed to the averaged latitude band (5-25S).

*We partially agree with this comment, but even the negative ones have an extent of 30 degrees, thus it is not really 'standing'. Though, compositing around Day 0 might be the reason for the anomalies near that day to be the most prominent.*

3. Figure 4 caption: The overlaid wind is at 850hPa or 700hPa. Why is it a different level from Figure 2?

*It is a mistake in Figure 4. Every wind quivers presented here are of 700 hPa. Thanks for catching the error, we have rectified it.*

4. L129: discussion 'ans' conclusion à 'and'

*Thanks. corrected*

5. L180: northeast-southwest –> northwest-southeast?

*The gyre is oriented from northwest-southeast, thus the orientation of the wavetrain is NE-SW.*

6. the low level (700hPa) circulations.

*Thanks, corrected.*